# Selective Properties of Mid-Infrared Tamm Phonon-Polaritons Emitter with Silicon Carbide-Based Structures

**DOI:** 10.3390/mi13060920

**Published:** 2022-06-10

**Authors:** Chengxuan Gong, Gaige Zheng

**Affiliations:** School of Physics and Optoelectronic Engineering, Nanjing University of Information Science and Technology, Nanjing 210044, China; 20201249507@nuist.edu.cn

**Keywords:** Tamm phonon-polaritons, selective emission, distributed Bragg reflector (DBR), rigorous coupled wave analysis (RCWA)

## Abstract

Electromagnetic (EM) absorbers and emitters have attracted much interest because of their versatile applications. A photonic heterostructure composed of silicon carbide (SiC) layer/germanium (Ge) cavity/distributed Bragg reflector (DBR) has been proposed. Selective emission properties have been investigated through rigorous coupled wave analysis (RCWA) method. The results illustrate that Tamm phonon-polaritons can be excited, and the magnetic field is partially centralized at the junction of Ge cavity and SiC film, aimed to improve the interactions of photon–phonon. The absorptivity/emissivity of the structure can be better optimized by controlling the coupling of surface modes with the incident wave. Near-unity absorption can be achieved through optimizing the SiC grating/Ge cavity/distributed Bragg reflector (DBR) multilayer structure with geometrical parameters of *d_s_* = 0.75 μm, *d_g_* = 0.7 μm, *d*_1_ = 1.25 μm and *d*_2_ = 0.75 μm, respectively. Physical mechanism of selective emission characteristics is deliberated. In addition, the simulation results demonstrate that the emitter desensitizes to the incidence angle and polarization state in the mid-infrared (MIR) range. This research ameliorates the function of the selective emitters, which provides more efficient design for SiC-based systems.

## 1. Introduction

Mid-infrared (MIR) electromagnetic (EM) absorbers/emitters have attracted great interest due to their intrinsic bountiful physical mechanisms and important practical applications in molecule fingerprinting [1], radiative cooling [2], as well as diagnostic tools in medical science [3]. Wavelength selective MIR absorber/emitter is absolutely essential for investigating the pertinent physics with optical components, such as sources, detectors, sensors and beam-steering devices [4,5,6,7,8,9,10,11].

Plasmonic structures show sharp and strong resonance because of the excitation of surface plasmon polariton (SPP) modes, which have been widely studied both theoretically and experimentally [12,13,14]. However, one of the existing problems is that they are highly sensitive to angle of incidence and polarization state. The other point is that the inherent optical loss related to the fast scattering lifetimes of plasmons impose unavoidable limitations on their appliance in the MIR range [15]. It is fortunate that polar dielectrics provide a chance to concurrently actualize sub-diffraction limitation and low loss in MIR through excitation of surface phonon polaritons (SPhPs). The SPhPs mode is generated by the coupling of the strong EM fields and phonons (vibrations in crystal lattices) of the polar crystal [16,17,18].

Quite recently, a hybrid structure that can support optical Tamm states (OTSs) has been proposed and investigated as a wavelength selective thermal emitter in the MIR range [19,20]. The OTS corresponds to the narrow-gap resonant mode with the localization of strong EM fields on the interface of a heterogeneous structure, resulting in an ultra-narrow emission peak. One type of OTS, named as Tamm plasmon polariton (TPP), can appear on the interface of a photonic crystal with a metal layer [21,22,23]. In contrast to SPP, the TPPs are standing waves that can be excited simultaneously under TE and TM polarizations. Moreover, the in-plane wave vector of TPPs is smaller than that of light in vacuum, leading to the direct optical excitation. Similar to TPPs, Tamm phonon-polaritons (TPhPs) have been predicted, which stem from the photonic band gap of the multi-porous multilayer and the coupling of phonon polaritons in silicon carbide (SiC) [24]. TPhPs could greatly improve the performance of narrow-band thermal emitters due to the significant enhancement of electric field and quality factor [24].

In this paper, we present a wavelength-tunable narrow-band emitter combined with SiC film or gratings at the top of a distributed Bragg reflector (DBR) structure that is composing of alternatively superposing germanium/zinc selenide (Ge/ZnSe) layers. TPhPs can be excited by tuning the coupling between the photonic crystal bandgap and the phonon polaritions excitations in the polar dielectric SiC film, and TPhPs can exist in both Transverse Electric (TE), Transverse Magnetic (TM) polarization states. Ge and ZnSe are selected to comprise the DBR since they offer a contrast of refractive index to achieve a broad photonic stopband, and a high reflectivity which is possible to be accomplished with only five layers of the overlapping films. In the numerical experiment, the refractive index of Ge is taken from Palik’s handbook of optical constants, *n_Ge_* = 4.0037 [25]. The optical characteristics of the ZnSe is taken from Ref. [26], the refractive index used is *n_ZnSe_* = 2.3950 for the ZnSe layer at normal incidence. Thickness of each DBR layer is set to one-quarter of the valid optical length of the central wavelength, the thickness of the Ge layer is 0.75 µm and the thickness of the ZnSe layer is 1.25 µm.

Additional Ge layer is introduced in the middle of the top SiC film/gratings and the bottom DBR, and the thickness of spacer layer (Ge nanocavity) is denoted as ds. The function of Ge spacer layer is equivalent to an optical cavity in the multilayer structure. The coupling between the TPhPs and cavity modes has been tuned by varying the thickness of optical cavity. The optical characteristics of resonance modes and its geometric correlation are systematically investigated. The induced absorptivity (i.e., emissivity) is demonstrated analytically by controlling the resonance of surface modes. It also demonstrated the polarization- and incident angle-independent emissivity.

## 2. Materials and Methods

The proposed multilayer structure can be denoted by silicon carbide (SiC) layer/germanium (Ge) cavity/distributed Bragg reflector (DBR) as shown in Figure 1. The multilayer structure is fabricated by chemical vapor deposition (CVD), which uses SiC as the substrate under low pressure vacuum conditions. The source gas is transported through the carrier gas into the reaction chamber, then it is rapidly pyrolyzed into an intermediate gas that diffuses onto the surface of the substrate, where it is adsorbed and undergoes a series of inhomogeneous reactions to produce the epitaxial layer, and it is characterized by Fourier Transform infrared spectroscopy (FTIR) microscopy. All SiC wafers in this paper are research grade and are manufactured by Tinker Heta Semiconductor Co., Ltd. in Beijing, China, with a purity of 99.95%. ZnSe is vapor deposited by high purity argon carrying zinc vapor into the reaction chamber with excess selenium vapor and is manufactured by Hangzhou Kaiyada Semiconductor Materials Co., Ltd. (Hangzhou, China) with a purity of 99.99%. Additionally, Ge is also made by Hangzhou Kaiyada Semiconductor Materials Co., Ltd., with purity up to 99.99%.

Polar dielectrics can stimulate surface phonon polaritons (SPhPs) mode, which is the result of coupling between the surface electromagnetic modes (photons) and the lattice vibrational modes (optical phonons) of polar materials. SPhPs is formed on the surface of polar dielectrics and inherently exhibit longer scattering lifetimes than SPPs, resulting in lower optical losses. The real part of the dielectric function of the SiC is negative within the Restsrahlen range [27,28]. The wavelength-relative dielectric constant of the SiC can be represented in the terms of the Drude−Lorentz model [29,30].
(1)$$\varepsilon_{SiC}=\varepsilon_{\infty }\frac{\omega^{2}-\omega_{LO}^{2}+ί\gamma \omega }{\omega^{2}-\omega_{LO}^{2}+ί\gamma \omega }$$
where ω*_LO_* and ω*_TO_* stand for the vertical and horizontal light phonon frequencies, with value of 972 cm^−1^ and 796 cm^−1^, respectively; ε_∞_ is intercalated as the high-frequency dielectric permittivity, *γ* is defined as the damping ratio caused by vibration and harmonic wave, and ω is the frequency of the incident light. ε_∞_ is selected for 6.5, and *γ* is chosen to be 3.75 cm^−1^ [31].

The absorption can be obtained by the formula of *A* = 1 − *R*, where *A* and *R* stand for absorptance and reflectance, respectively. On the basis of Kirchhoff’s law, the thermal emission on the plane surface is equivalent to its absorption. This means that one can calculate the reflectance (*R*) to simulate thermal emission indirectly through rigorous coupled wave analysis (RCWA) calculation [32,33] and also through theoretical models [34,35]. The reflectance of the DBR is different for odd and even numbers of stacks, and the reflectance of an odd number of layers can be calculated by the following equation.
(2)$$R_{2N+1}={\left|\frac{1-\frac{n_{1}^{2}}{n_{0}n_{3}}{\left(\frac{n_{1}}{n_{2}}\right)}^{2N}}{1+\frac{n_{1}^{2}}{n_{0}n_{3}}{\left(\frac{n_{1}}{n_{2}}\right)}^{2N}}\right|}^{2}$$

For an even number of layers, the reflectance can be obtained by the following equation.
(3)$$R_{2N}={\left|\frac{1-\frac{n_{3}}{n_{0}}{\left(\frac{n_{1}}{n_{2}}\right)}^{2N}}{1+\frac{n_{3}}{n_{0}}{\left(\frac{n_{1}}{n_{2}}\right)}^{2N}}\right|}^{2}$$
where *n*_1_ and *n*_2_ are the refractive indices of the first and second layers in the DBR, *n*_0_ and *n*_3_ are the refractive indices of the incident and emitted media, respectively, and *N* is the number of periods.

## 3. Results

### 3.1. Selective Emission with Multilayer Planar Thin Films

Emitter with narrower bandwidth owns great potential for applications as optical filters, detectors and biosensors. Figure 2 illustrates the emission spectra and reflection spectra with different numbers of DBR pairs (*N*) under normal incidence. The emitter with 4 pairs provides much narrower emission spectra compared to the emitter with 3 DBR pairs, while the intensity of the emission peak increases from 70% to 91%. With the increase of *N*, the intensity of resonance reflectivity decreases slowly while the resistance band of DBR reflection spectrum decreases. The space of DBR becomes larger, the energy confined around the DBR decreases with the result that more energy is radiated out, and the emission spectrum and reflection spectrum gradually blue-shifts. The intensity of resonance emissivity will increase to nearly 100% when *N* = 7. Nevertheless, an optimal number of DBR pairs usually exists for consideration of the thickness of the whole structure. Near perfect narrowband absorption (emission) at *λ* = 12.8 μm can be realized when *N* = 5. Additionally, *N* will be fixed at 5 in the following study with the aim of investigate the influence of other geometric structure parameters on the performance.

The function of Ge spacer layer is equivalent to an optical cavity in the multilayer structure. The emissivity as a function of wavelength and *d*_s_ is plotted in contour Figure 3, the emission can be significantly changed by adjusting *d*_s_. A splitting effect can be observed, and perfect absorption is achieved with *d_s_* = 0.75 μm. The coupling between the TPhPs and cavity modes has been tuned by varying *d*_s_. At different values of *d*_s_, there is an anticrossing between the two modes, leading to different Rabi-like splitting values for the modes.

To check the tunability of the resonance, effects of structural parameters on the emission spectrum is evaluated. Perfect absorption can be achieved by optimizing *t*, *d*_1_ and *d*_2_. The absorption will be influenced by *t*, and the resonance emission red-shifts when *t* varies from 0.3 μm through 0.6 μm, as seen from Figure 4a. The maximum peak absorptance decreases with increasing *t*. The optimal value of *t* is chosen as 0.45 μm. As seen from Figure 4b, the absorptance band presents a red-shift as *d*_1_ increases. Moreover, the resonance emission meets a splitting with increasing *d*_2_, as depicted in Figure 4c. The position and intensity of the absorption peak can be adjusted with the changing of the incident angle, which is extremely important for MIR emission. Figure 5a,b show the relationship between the emission spectra and the incident angle under TE and TM polarizations. As on can see in Figure 5a, when incident angle varies from 0° to 40°, the absorption intensity is very stable with a narrowing bandwidth. The absorption intensity reduces rapidly with the incident angle over 60°. The emission spectra under TM polarization is illustrated in Figure 5b. Intuitively, a flat and near-perfect absorption band appears in the vicinity of 12.8 μm. In order to better understand the response, Figure 5c,d describe the angular distribution of thermal emission of the structure at different wavelength. It is shown that a wide emission in the normal direction and the angular width can reach 50°.

To prove the formation of OTS, the field intensity distribution (|H_y_|) at the peak wavelength is calculated and depicted in Figure 6. The dotted lines show the boundaries of different layers. As shown in illustration, the magnetic field is mostly restricted around the boundary, and the strength is enhanced by more than 80 times. Both the absorption position within the bandgap and strongly confined magnetic field demonstrate the presence of OTS at the top interface, which can administer to the strong enhancement of light–matter interactions.

### 3.2. Selective Resonance Response with SiC Gratings

It has been demonstrated that gratings can be stacked on top of DBR to enhance the polarization-correlation emission rate, selective resonance and high radiation directional property [33,36]. Figure 7 shows a diagrammatic drawing of the photonic Tamm structure, which is composed of a one-dimensional (1D) periodic SiC grating deposited onto a ZnSe/Ge-based DBR (consisting of 5 pairs of ZnSe/Ge alternances with *λ*/4 layer thickness and center wavelength *λ*_Bragg_ = 12 μm) with an additional Ge spacer layer. The structure is defined in terms of period *p*, grating ridge width *w*, grating thickness *d*_g_, and filling factor *f* = *w*/*p*. 𝜃 is the angle of the incident beam. The multi-layer architecture could be manufactured by means of chemical vapor deposition, and the one-dimensional lattice of SiC gratings could be patterned with electron beam lithography. 

Variations of the structural parameters will affect the spectra characteristics of the resonance. Figure 8a displays the calculated results through varying the grating period (*p*) between 1.5 and 2.5 μm, with filling factor *f* = 0.5, and the grating height (*d*_g_) = 700 nm. The other parameters are the same as used in Figure 2. It is clearly that the maximum emissivity can be achieved with *p* = 2 μm, when it is in resonance. As *p* increases further, the grating produces plane waves propagating at the interface, and the interference pattern produced by the plane waves produces a maximum in the field superimposed on the grating structure. This produces a Joule effect, increasing losses and thus leading to a reduction in emissivity. The similar effects have been observed on the change of *f* and *d*_g_, as illustrated in Figure 8b,c.

Figure 9a,b present the wavelength and angle-dependent emission of the SiC-based structure with *p* = 2 μm, *f* = 0.5, *d*_g_ = 0.7 μm and *d_s_* = 0.75 μm. Near-unity absorption for both TE and TM polarization has been achieved for angles less than 80°. For angles larger than 80°, the intensity of resonance decreases significantly for TE polarization, while can be perfectly kept under TM polarization. Figure 9c,d describes the angular distribution of the emission for two different wavelengths. As expected from the results, the proposed design shows a wide emission in the normal direction and the angular width reaches 80°. In both polarization states, narrowband emission can be maintained at various incident angles. Compared with the structure based on SiC planar film, the angular width of the emission increases to 80° and the intensity increases from 96.74% to 99.54%. Therefore, gratings can be stacked to maintain high emission for all angles and both polarization states.

Figure 10 demonstrates the magnetic field profile |H_y_| of the surface modes under resonance condition (*λ* = 12.36 μm) at *θ* = 0°. The strong magnetic field is centralized within the grooves of the SiC grating. The analyte can penetrate into the grooves readily, thus gathers over the grating. The reduction of analyte concentration within the grooves causes small disturbance of the refractive index that demonstrates the potential utilization of the presented structure as the sensor.

## 4. Conclusions

We propose a wavelength tunable and narrow-band emitter by means of combining SiC film and gratings at the top of the classical DBR structure that is composed of alternative germanium/zinc selenide (Ge/ZnSe) layers. The optical property of resonance modes and its geometric dependence on the hybrid structure are systematically investigated. The intensified absorptivity (i.e., emissivity) of the designed wavelength is demonstrated analytically. By controlling the coupling of surface modes with the incident wave, we illustrate in detail the intensified absorptivity (i.e., emissivity) of the designed wavelength. The results also show that the emitter demonstrates emissivity insensitivity to polarization and incident angle. We find that the structure can achieve perfect thermal emission (absorption) in the MIR range through the use of the Tamm state as well as its local enhancement features of magnetic field. This structure can provide important guidance for the design of new thermal emitters for energy conversion and other applications.

## Figures and Tables

**Figure 1 micromachines-13-00920-f001:**
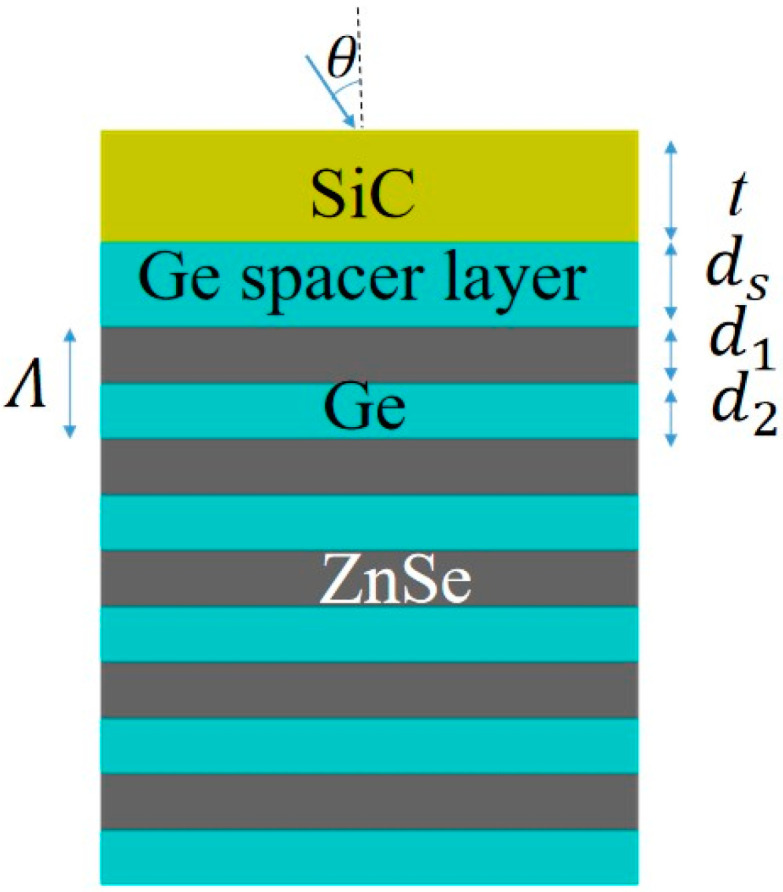
Cross-section view of the thermal emitter based on SiC film with DBR reflector.

**Figure 2 micromachines-13-00920-f002:**
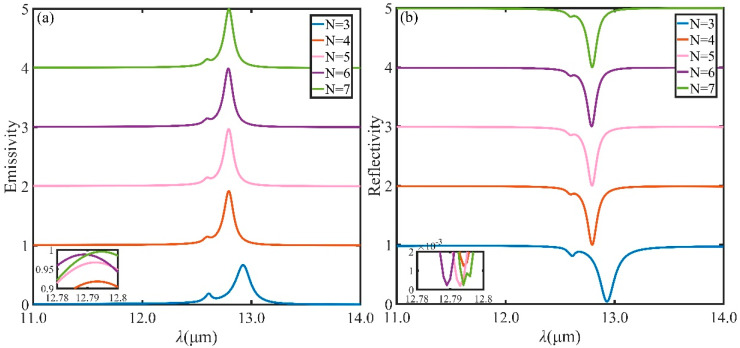
(**a**) Emissivity at normal incidence for TM polarization with different number of distributed Bragg reflector (DBR) pairs (*N*). (**b**) Reflectivity at normal incidence for TM polarization with different number of distributed Bragg reflector (DBR) pairs (*N*). The other parameters are chosen as *t* = 0.45 μm, *d*_s_ = 0.75 μm, *d*_1_ = 1.25 μm and *d*_2_ = 0.75 μm.

**Figure 3 micromachines-13-00920-f003:**
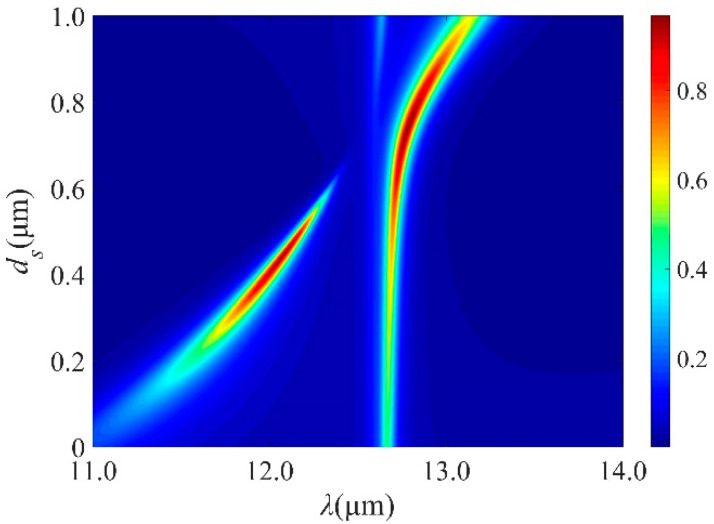
Emission spectra of the design with different *d_s_* under normal incidence for TM polarization with 5 DBR pairs when *t* = 0.45 μm, *d*_1_ = 1.25 μm and *d*_2_ = 0.75 μm.

**Figure 4 micromachines-13-00920-f004:**
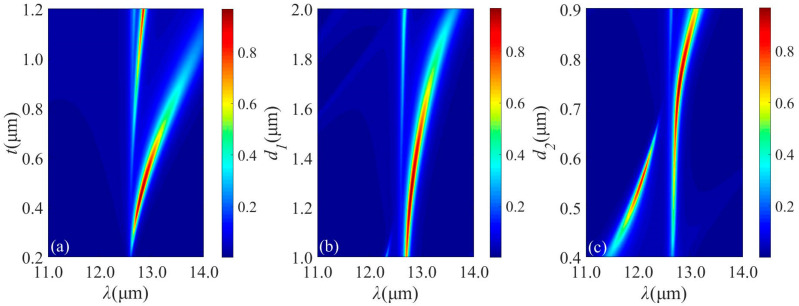
(**a**) Emission spectra of SiC film with different *t* under normal incidence for TM polarization when *d*_s_ = 0.75 μm, *d*_1_ = 1.25 μm and *d*_2_ = 0.75 μm with 5 DBR pairs. (**b**) Emission spectra with different *d*_1_ under normal incidence for TM polarization when *t* = 0.45 μm, *d*_s_ = 0.75 μm and *d*_2_ = 0.75 μm with 5 DBR pairs. (**c**) Emission spectra with different *d*_2_ under normal incidence for TM polarization when *t* = 0.45 μm, *d*_s_ = 0.75 μm and *d*_1_ = 1.25 μm with 5 DBR pairs.

**Figure 5 micromachines-13-00920-f005:**
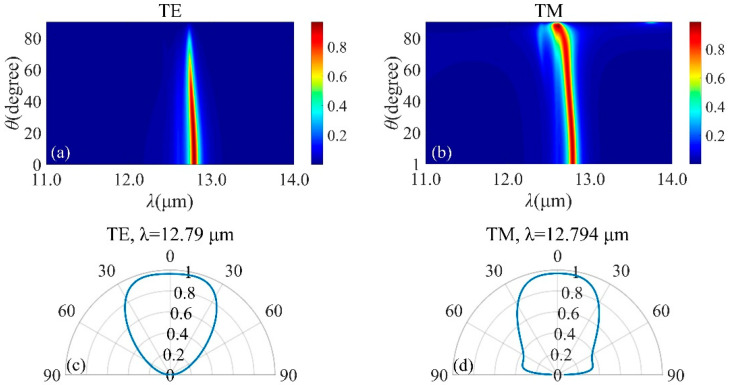
Emission spectra of SiC−based structure with 5 DBR pairs when *t* = 0.45 μm, *d*_s_ = 0.75 μm, *d*_1_ = 1.25 μm and *d*_2_ = 0.75 μm for TE− (**a**) and TM− (**b**) polarization. Angle distribution of the emission with 5 DBR pairs with *t* = 0.45 μm, *d*_s_ = 0.75 μm, *d*_1_ = 1.25 μm and *d*_2_ = 0.75 μm at λ = 12.79 μm and λ = 12.794 μm are illustrated for TE− (**c**) and TM− (**d**) polarizations, severally.

**Figure 6 micromachines-13-00920-f006:**
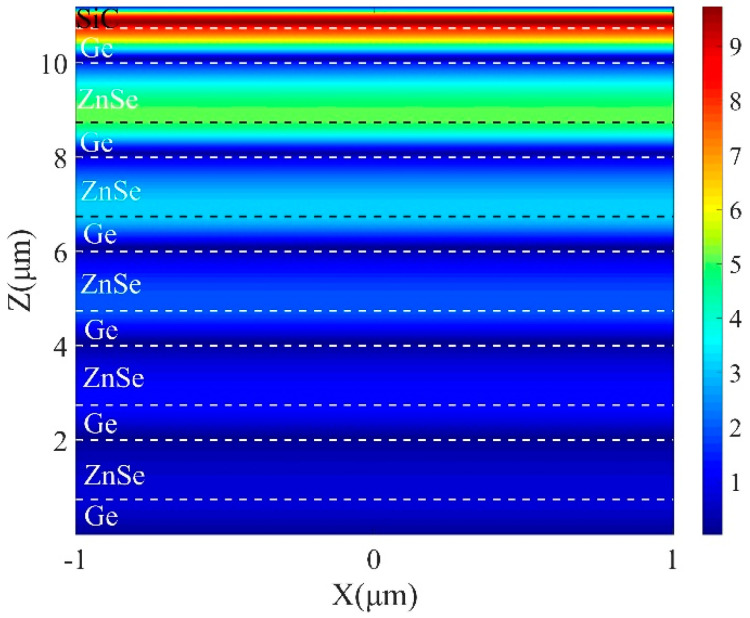
Magnetic field distribution of TM polarization in x−z plane with 5 DBR pairs at λ = 12.8 μm when *t* = 0.45 μm *d_s_* = 0.75 μm, *d*_1_ = 1.25 μm and *d*_2_ = 0.75 μm.

**Figure 7 micromachines-13-00920-f007:**
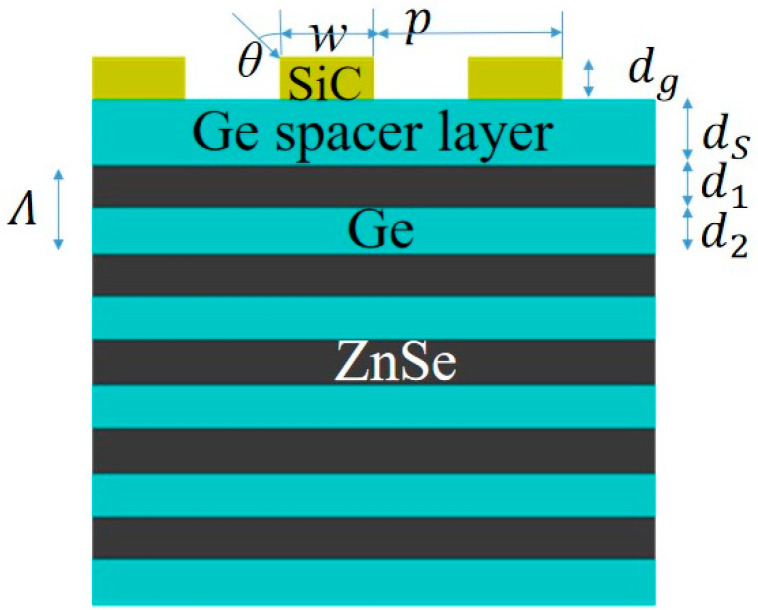
Cross-section view of the thermal emitter based on SiC grating with DBR reflector.

**Figure 8 micromachines-13-00920-f008:**
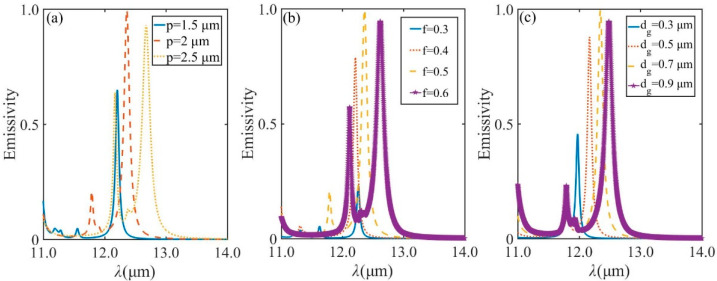
(**a**) Emission spectra under normal incidence for TM polarization with 5 DBR pairs when *f* = 0.5 and *d*_g_ = 0.7 μm with grating period *p* of 1.5 μm, 2 μm, 2.5 μm, respectively. (**b**) Emission spectra under normal incidence for TM polarization with 5 DBR pairs with *p* = 2 μm and *d*_g_ = 0.7 μm when *f* = 0.3, *f* = 0.4, *f* = 0.5, *f* = 0.6, respectively. (**c**) Emission spectra under normal incidence for TM polarization with 5 DBR pairs with *p* = 2 μm and *f* = 0.5 with grating depth *d*_g_ =0.3 μm, *d*_g_ = 0.5 μm, *d*_g_ = 0.7 μm, *d*_g_ = 0.9 μm, respectively.

**Figure 9 micromachines-13-00920-f009:**
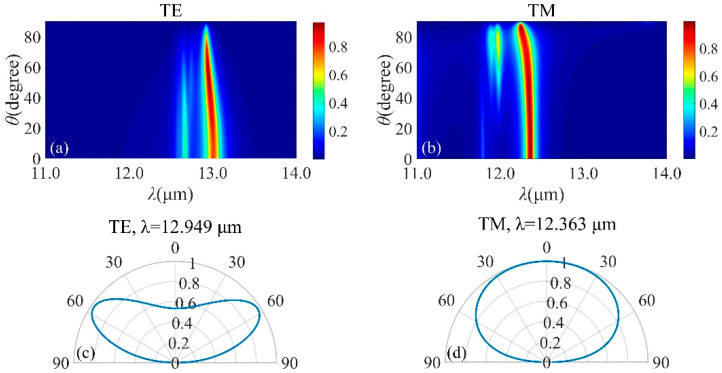
Emission spectra of SiC gratings−based structure with 5 DBR pairs when *p* = 2 μm, *f* = 0.5 and *d*_g_ = 0.7 μm for TE− (**a**) and TM− (**b**) polarization. Angle distributions of the emission at *λ* = 12.949 μm and *λ* = 12.363 μm are shown for TE− (**c**) and TM− (**d**) polarizations, respectively.

**Figure 10 micromachines-13-00920-f010:**
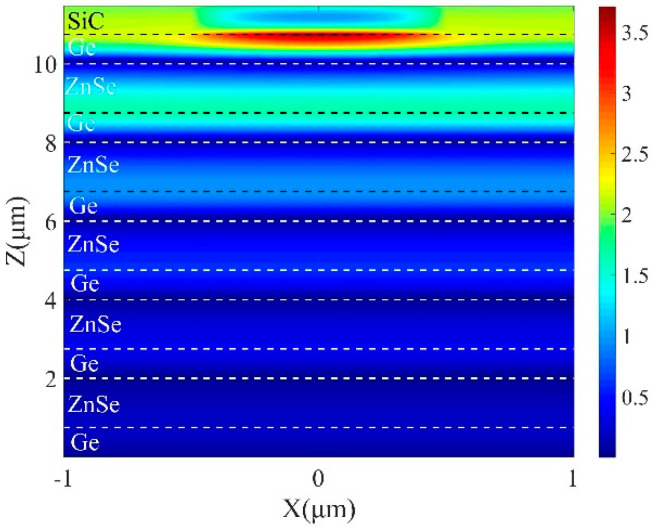
Magnetic field distribution of TM polarization in x−z plane with 5 DBR pairs with *p* = 2 μm, *f* = 0.5 and *d*_g_ = 0.7 μm at *λ* = 12.35 μm.

## Data Availability

Not applicable.

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
