# Peer review of "Selective Properties of Mid-Infrared Tamm Phonon-Polaritons Emitter with Silicon Carbide-Based Structures"

_micromachines, 2022, doi:10.3390/mi13060920_

Round 1
Reviewer 1 Report
The paper presents important work regarding selective mid-infrared photons-phonons interactions for absorber/emitters applications. Different parameters are tuned to get optimum results. The optical characteristics of resonance modes and their geometric correlation is studied. Overall, the paper is well-written, however, the following points must be incorporated to make the article understandable and interesting for the readers;
Abstract section
1. The abstract should contain important findings of the study, i.e., the optimization conditions for better absorption/emission with the corresponding values of t, ds, d1 and d2 etc.
Introduction section
2. Introduction section should include the photonic, optical, refractive indices and structural information of SiC, Ge, and ZnSe in order to justify their use in this work.
3. Please include details that what is the role of additional Ge spacer layer in the middle of the top SiC film and the bottom DBR.
4. Line no. 46 page#2, TE and TM should be first defined
Materials and Methods section
5. This section must include the experimental details how the silicon carbide (SiC) layer/germanium (Ge) cavity/ distributed Bragg reflector (DBR) structure have been developed and under what conditions as well as mention proper models of the instruments used for fabrication of the structure and characterizations.
6. From where, manufacturer/company, the materials are obtained and what is their purity.
7. How polar dielectrics stimulate surface phonon polaritons (SPhPs) mode, include mechanism.
8. All the parameters in equation (1) should be properly defined.
9. How the values of ε∞ and γ are chosen? Quote with proper reference
Results section
10. In figure 2, the maximum number of DBR pairs (N) is shown to be N=3,4,5, however, in line no. 96, it has been stated that at N=7 the intensity of resonance emissivity will increase to 100%. How this statement is validated?
11. On page no. 6, “It is clearly that the maximum emissivity can be achieved with p = 2 µm, further increasing of p will lead to a decrease of the intensity.” Give detailed mechanism to clarify this statement.

Author Response
Dear reviewer,
We have studied the valuable comments from you carefuly, and tried our best to revise the manuscript. The point to point responds to the reviewer’s comments are listed as in attachment.

Reviewer 2 Report
Please see the attachment.

Author Response

(The authors gave the same response as above.)

Round 2
Reviewer 1 Report
The authors have much improved the manuscript in the revised version.
However, in the Introduction section, the optical properties and refractive indices as well as structural information (crystal structures etc.) of SiC, Ge, and ZnSe is still missing. I suggest to include this data to be readily available to the readers instead of giving reference.
After the above changes, I recommend the article for publication in Micromachines.
Author Response

(The authors gave the same response as above.)
